# Exploring emergency department 4-hour target performance and cancelled elective operations: a regression analysis of routinely collected and openly reported NHS trust data

Brad Keogh,[1] David Culliford,[1] Richard Guerrero-Ludueña,[2] Thomas Monks[1]

## ABSTRACT

**Objective** To quantify the effect of intrahospital patient flow on emergency department (ED) performance targets and indicate if the expectations set by the National Health Service (NHS) England 5-year forward review are realistic in returning emergency services to previous performance levels.

**Design** Linear regression analysis of routinely reported trust activity and performance data using a series of cross-sectional studies.

**Setting** NHS trusts in England submitting routine nationally reported measures to NHS England.

**Participants** 142 acute non-specialist trusts operating in England between 2012 and 2016.

**Main outcome measures** The primary outcome measures were proportion of 4-hour waiting time breaches and cancelled elective operations.

**Methods** Univariate and multivariate linear regression models were used to show relationships between the outcome measures and various measures of trust activity including empty day beds, empty night beds, day bed to night bed ratio, ED conversion ratio and delayed transfers of care.

**Results** Univariate regression results using the outcome of 4-hour breaches showed clear relationships with empty night beds and ED conversion ratio between 2012 and 2016. The day bed to night bed ratio showed an increasing ability to explain variation in performance between 2015 and 2016. Delayed transfers of care showed little evidence of an association. Multivariate model results indicated that the ability of patient flow variables to explain 4-hour target performance had reduced between 2012 and 2016 (19% to 12%), and had increased in explaining cancelled elective operations (7% to 17%).

**Conclusions** The flow of patients through trusts is shown to influence ED performance; however, performance has become less explainable by intratrust patient flow between 2012 and 2016. Some commonly stated explanatory factors such as delayed transfers of care showed limited evidence of being related. The results indicate some of the measures proposed by NHS England to reduce pressure on EDs may not have the desired impact on returning services to previous performance levels.

[1]NIHR CLAHRC Wessex, University of Southampton, Southampton, UK
[2]Wessex AHSN CIS, University of Southampton, Southampton, UK

**Correspondence to**
Dr Brad Keogh;
brad.keogh@soton.ac.uk

## Strengths and limitations of this study

► This study is the first to examine in detail the in-hospital factors that influence emergency department 4-hour performance.

► We analyse the change in the importance of common explanatory theories of hospital flow bottlenecks across 5 years using recent openly published data.

► There are potentially some data quality issues around the variables used in the analysis, which may influence the conclusions.

► A relatively simple methodology was used to ensure transparency of the study, it is possible that some statistical inference from the data is lost that more complex methodologies might reveal.

► Future work involving the use of more complex statistical methodologies and the investigation of the relative importance of patient flows within trusts, population factors and data quality would be of use to further understand the different pressures trusts face and aid in the targeting of service reconfigurations.

## INTRODUCTION
### Background

It is widely reported that pressures on acute National Health Service (NHS) trusts across England have been steadily increasing in recent years.[1–5] This is often reported in the media as rising numbers of breaches of the 4-hour target, which is calculated as the percentage of patients being treated within 4 hours of arriving at an emergency department (ED). Concerns over the number of cancelled elective operations at trusts, as a result of increasing emergency pressures, have also been highlighted.[5] There is increasing pressure for NHS services to return to a 95% adherence of the 4-hour target as part of the NHS England 'Next steps on the NHS 5 year forward view',[6] which has

received some criticism.[7] Deliverables have been set, which include looking to increase 'front door streaming', improving patient flow and reducing delayed transfers of care. There is currently uncertainty around the impact these interventions will have and if they are likely to result in a return to the performance targets expected.

There are several analyses that aim to understand the causes for ED performance decline and increasing pressure on acute services in England over recent years. Some analyses and commentary have suggested that high rates of bed occupancy and delayed transfers of care within trusts could be increasing the pressure on acute services, and in some cases may lead to increased waiting times in EDs.[3 8–11] However, there is currently limited peer-reviewed statistical evidence showing the relationship between these factors and routinely collected measures of pressure on acute trusts. One of the few peer-reviewed works that has been published[12] examined data over a single 2-week period in 2002 and used linear regression analysis to show a relationship between hospital bed occupancy and 4-hour target performance. The study is univariate, however, and is limited to only a small time window, hence it only provides a limited amount of understanding of the contributing factors to 4-hour target performance. Other quantitative studies investigating the cases and consequences around waiting times in EDs[13–22] have looked at data that are not openly published and often pertain only to a small number or single hospital not necessarily located in the UK. Many of these studies also only look at the ED in isolation. These studies have limited scope in explaining the relative importance of different factors affecting growing emergency pressures on hospitals in England.

A systematic quantitative study is required, which looks at the patient flow factors across trusts to understand the factors that trusts can modify, which could improve performance and care. Greater understanding of these factors may allow appropriate targeting of resources tailored to trusts, rather than a suite of measures that are expected to be implemented across all providers. The aim of this work is to investigate the relative impact of the commonly highlighted variables: bed occupancy, delayed transfers of care and other routinely measured operational factors, on the 4-hour target as well as on cancelled elective operations across Acute trusts in England. This will provide evidence of the relative importance of each of these factors as well as how these dependencies have changed over time. It will also demonstrate how routinely collected and openly reported data can be used in a statistically robust way to understand more about how pressures on NHS services are changing.

### Simplified high-level system flow
Figure 1 shows a simplified hospital trust system and the patient flows through it. Patients attending ED will either be admitted as an inpatient to the trust or will be discharged from the ED. Patients may only be admitted from ED to the inpatient provision if there is space available (usually in the form of a bed). Space will only become available as inpatients are discharged or transferred from the trust to home, social care provision or another trust. Additionally, there is pressure to admit from patients undergoing elective (non-emergency) procedures. Any of these patient flows could be a bottleneck, which can result in a deterioration of trust performance measures.

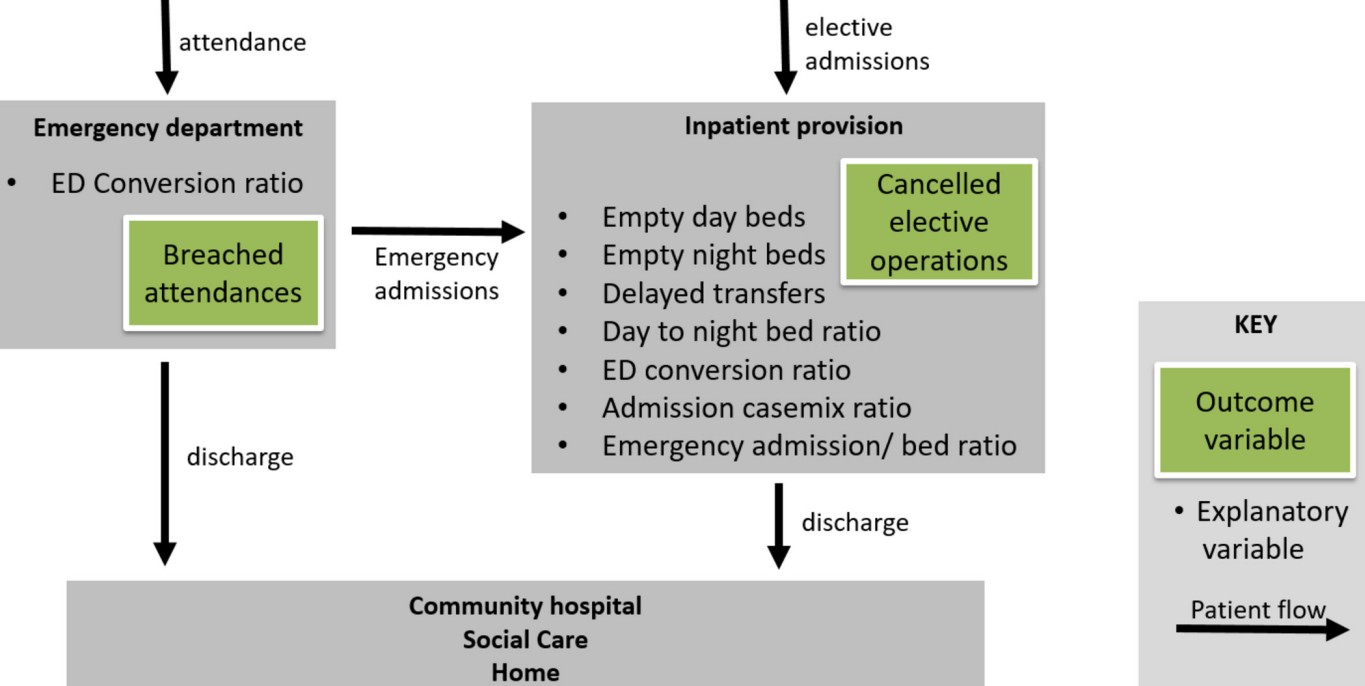

**Figure 1** Simplified trust system broadly illustrating patient flow. ED, emergency department.

Two measures of the pressure a trust is facing are breached attendances and cancelled elective operations; these are commonly thought to be a measure of pressure on EDs and inpatient provision, respectively. There are several factors, related to patient flow, that may provide insight into the pressures that trusts are currently facing. Those which will be investigated in this study are related to bed occupancy, which is reported both in day and night beds; day bed to night bed ratio, indicating the split between bed types in a trust; delayed transfers of care; conversion ratio, the proportion of patients attending ED who are admitted; casemix ratio, indicating the split between emergency and elective admissions; and the number of admissions per bed within a trust. Figure 1 highlights these measures around the area they are likely to affect the hospital system most acutely.

### Internal workings of an ED

The efficiency of internal ED microflow has been analysed extensively using modelling and simulation.[23–25] This, for example, has explored the use of fast tracks for minor injuries or patient 'streaming' (now commonplace in England),[26] prioritisation by acuity,[17] and workforce scheduling and resourcing.[27] Processes of patient flow through ED and into trusts have also been shown to vary considerably between sites.[17] However, our unit of analysis is the hospital and its effects on 4-hour performance.

### METHODS
### Data collation

A data set was collated from published open data hosted on the NHS England Statistics website.[28] These data sets included data for all NHS trusts in England as well as minor injury units and walk-in centres. At the time of collation, data were available from quarter 2, 2011 until quarter 4, 2016 (calendar year) for the statistical reports entitled:
► Emergency department attendances and emergency admissions.
► Bed availability and occupancy.
► Cancelled elective operations.
► Delayed transfers of care.
► Hospital activity.

The data source is created from each NHS organisation routinely reporting their own counts of activity, which NHS England collate and publish. This is a separate source from the Secondary Uses Service or Hospital Episodes Statistics. Full definitions of the indicators and the rules for submission by the providers are available from NHS England.[28]

### Trust filtering

The collated dataset was filtered to only include NHS trusts in England defined as Small, Medium, Large Acute and Teaching. These trusts are the organisations who have seen the greatest reduction in the 4-hour target. Mental health trusts, acute specialist trusts, walk-in centres, practices, health centres, out-of-hours services and treatment centres were all excluded from the analysis. The definitions for NHS trust types are available online.[29]

### Study variables

The collated data contained counts of events for each NHS trust in England, which were converted into a proportion or a ratio using an appropriate denominator specific to the same trust for each time-period. For example, the 'number of attendances in ED lasting greater than 4 hours' was divided by 'total number of ED attendances' of the period at that trust, and the 'number of beds occupied' were divided by 'total number of beds'. This created variables, which allowed useful comparisons between trusts of different sizes and activity levels. A summary of the variables investigated in this study are included in table 1, along with information on how they were calculated and a description of how they should be interpreted. The variables in this study were created from aggregating quarterly data for each year between 2012 and 2016 and relate to those in figure 1. The raw collated quarterly count data as well as the python code to produce the year-aggregated variable data used in this study have also been made available.[30]

### Variable distributions and transformation

Some variables were transformed to provide a normal distribution for regression model fitting. After conversion into proportions/ratios, non-normal variables were transformed using a natural log function (cancelled electives, empty night beds, admission casemix ratio). One variable (empty day beds) contained zero values and hence a log transformation was not appropriate. This variable was categorised. Bin edges were determined to provide approximately equal numbers of samples in each group. Where this was conducted, bin sizes were created based on data across all years of study in order to provide consistent transformation.

### Bias

Some missing data were found where trusts had not submitted data. For each year studied, no more than 4% of trusts were found to have missing data for at least one variable; therefore, the maximum percentage of missing data points for any regression was less than 4%. Our initial protocol intended to blind the variables throughout the analysis; however, variables were unblinded part way through the study as it was decided that greater contextual understanding of the problem was required to fully develop the analysis.

Within Trusts, the reported activity is split into types 1, 2 and 3. Type 2 and type 3 EDs are defined by NHS England as Minor Injury Units, Eye Casualties, Urgent Care Centres and Walk-In Centres. At some trusts, these services are co-located at the same hospital site and so these could not be excluded from the analysis as the attendances contribute to hospital patient flow. It is possible that some 4-hour target variation across Trusts may be due

**Table 1** Variables included in the study

| Variables | Type | Numerator | Denominator | Units | Transformation applied? | Interpretation |
|---|---|---|---|---|---|---|
| Breached attendances | Outcome | Number of ED attendances greater than 4 hours | Number of ED attendances | – | – | Proportion of ED attendances waiting >4 hours |
| Cancelled electives | Outcome | Number of cancelled elective operations | Number of elective admissions | Operations per admission | Log | Ratio of cancelled elective operations to elective admissions. In absence of the number of planned elective operations, this is the most suitable denominator |
| Empty day beds | Explanatory | Number of unoccupied day beds | Number of day beds | – | Categorised (5) | Ratio of unoccupied day beds to total number of day beds |
| Empty night beds | Explanatory | Number of unoccupied night beds | Number of night beds | – | Log | Ratio of unoccupied night beds to total number of night beds |
| Delayed transfers | Explanatory | Number of bed days taken by delayed transfers | Number of night beds | 10 bed-days | – | The number of bed-days lost to delayed transfers for each night bed at a trust, over the course of a year |
| Day bed:night bed ratio | Explanatory | Number of day beds | Number of night beds | – | – | Ratio of total day beds to total night beds |
| ED conversion ratio | Explanatory | Number of emergency admissions via ED | Number of attendances at ED | – | – | Ratio of ED admissions to attendances. Often commonly referred to as 'conversion ratio' |
| Admission casemix ratio | Explanatory | Number of non-elective (emergency) admissions | Number of elective admissions | – | Log | Proportion of admissions that are emergency or ratio of emergency to elective admissions |
| Emergency admission/bed ratio | Explanatory | Number of non-elective (emergency) admissions | Number of day and night beds | 10 admissions | – | Number of emergency admissions per bed over the course of a year |

ED, emergency department.

to larger volumes of patients attending types 2 and 3 units for minor injuries. As Trusts are often measured by their 4-hour target performance based on all ED types, and the focus of this work is on patient flow across Trusts, the analysis conducted in this paper includes attendances at all types of department within each Trust.

### Statistical methods
Univariate ordinary least squares linear regression was conducted using breached attendances as the outcome variable against each of the explanatory variables. To ascertain how the importance of each variable has changed over time, the regression analysis was performed for each year separately, in a series of cross-sectional studies. As this method could present a statistical problem of multiple comparisons, undue emphasis was not placed on statistical significance tests. Results are presented as an exploratory study, showing the regression coefficients, associated CIs and coefficient of determination values in each case. Only consistent associations that are of clinical importance are highlighted in the discussion. Multivariate regression was

also performed to ascertain the relative importance of each predictor variable on breached attendances when combined into a single model. A model containing all predictor variables that have been highlighted of clinical importance, and those that showed considerable association strength to the outcome variable in the univariate regression analysis, was created to provide some understanding of the interaction between predictor variables. The univariate and multivariate models' residuals were checked visually for normality and homogeneity. Influential outliers with high leverage were also investigated using Cook's distance, but not removed. Abnormalities are reported in the Results section. The same method of analysis was repeated using cancelled elective operations as the explanatory variable. Examples of the plots used to check the models are provided in online supplementary appendix A.

All the analysis conducted in this work was completed with the python language (V.3.6.0; www.python.org) using the Statsmodels (V.0.8.0; www.statsmodels.org) and

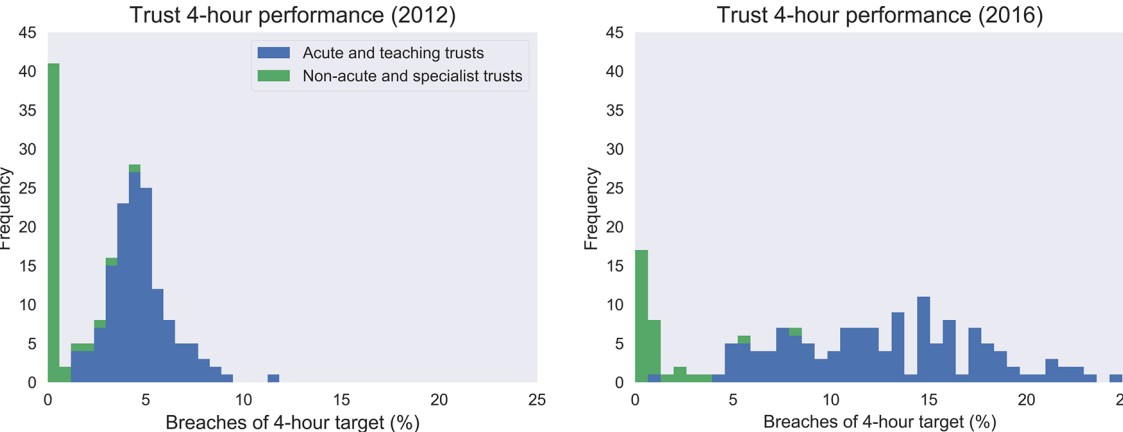

**Figure 2** Distribution of 4-hour target breaches by organisation type for 2012 and 2016.

Pandas (V.0.19.2; pandas.pydata.org) libraries. This was done in an Anaconda environment (www.anaconda.com) using Jupyter notebooks (www.jupyter.org).

### Patient and public involvement

There was no patient or public involvement in this study.

### RESULTS

In 2012, there were 254 organisations in England reporting at least one of the variables investigated in this study (226 in 2016). Figure 2 shows the distribution of 4-hour target breaches by organisation type. The acute and teaching trusts are observed to have higher proportions of breaches, with increases between 2012 and 2016. Most non-acute and specialist trusts still conform to the 4-hour target in 2016. Hence, the former group were the focus of this study. The number of trusts reduced to 142 when applying the criteria of the organisation type for the study (135 in 2016). Missing data reduced the number of trusts for some variables to 136 (131 in 2016).

### How variables have changed over time

Table 2 quantifies changes of each variable between 2012 and 2016. Between 2012 and 2016, the median values of breached attendances and cancelled elective operations have increased. The spread of the values has also increased in the time period in both cases, however more noticeably by over three times in the case of 'breached attendances'. The most noticeable increases occurred between 2015 and 2016 in both cases. There has been a decline in the proportion of empty night beds between 2015 and 2016 (it should be noted that this variable is the reverse of occupancy; 'empty beds'=1−'occupancy'). This reflects the historical reduction in night beds, which has been noted elsewhere.[10] Between 2012 and 2016, there has been a noticeable increase in delayed transfers and emergency admission to bed ratios across trusts. Median bed-days and IQR lost due to delayed transfers has doubled over the period. Less prominent increases are observed in empty day beds, day bed:night bed ratio and admission casemix ratio over this period. Although there are some fluctuations in ED conversion ratio, there is no overall change across the period.

### Breached attendances: univariate regression analysis

Table 3 gives the results of the univariate regression models using breached attendances as the outcome variable. Night-bed emptiness and ED conversion ratio showed respectively positive and negative associations, with breached attendances, consistently for each year of study. For night-bed emptiness in 2012, the $R^2$ value 0.10 is of similar magnitude to a previous study,[12] however in subsequent years is observed to be reduced. The day bed:night bed ratio showed a negative relationship with trust breaches, which increased in strength during 2015–2016; it also shows an increasing ability to account for breaches (increasing $R^2$ value between 2014 and 2016). The delayed transfers variable showed little evidence of a linear association with breached attendances between 2012 and 2015.

### Breached attendances: multivariate regression analysis

Table 4 shows the results of the multivariate regression for 2012 and 2016. In 2012, empty night beds and ED conversion ratio variables were both statistically significant and the model was able to explain 19% of the variation in the breached attendances variable. In 2016, the results of the multivariate model show only day bed:night bed ratio to be statistically significant in predicting breached attendances. The $R^2$ value indicates that only 12% of the variation in breached attendances can be accounted for with the parameters investigated in this study in 2016. When applying the multivariate model for other years, a reduction in the importance of empty night beds and ED conversion ratio, and an increase in importance of day bed:night bed ratio were steadily observed between 2012 and 2016.

### Cancelled elective operations: univariate regression analysis

Table 5 gives the results of the univariate regression models using cancelled elective operations as the outcome variable. ED conversion ratio, admission casemix ratio and delayed transfers showed a positive relationship with

**Table 2** Changes in distribution of explanatory variables between 2012 and 2016

| Variable | | 2012 | 2013 | 2014 | 2015 | 2016 |
|---|---|---|---|---|---|---|
| Breached attendances | Median | 0.05 | 0.05 | 0.06 | 0.07 | 0.13 |
| | IQR | 0.02 | 0.02 | 0.03 | 0.06 | 0.07 |
| | 5%–95% | 0.05 | 0.07 | 0.10 | 0.11 | 0.16 |
| Cancelled electives | Median | 0.012 | 0.013 | 0.013 | 0.013 | 0.015 |
| | IQR | 0.007 | 0.008 | 0.010 | 0.010 | 0.011 |
| | 5%–95% | 0.019 | 0.019 | 0.020 | 0.021 | 0.026 |
| Empty day beds | Median | 0.10 | 0.10 | 0.09 | 0.11 | 0.11 |
| | IQR | 0.17 | 0.17 | 0.17 | 0.17 | 0.21 |
| | 5%–95% | 0.38 | 0.37 | 0.43 | 0.40 | 0.37 |
| Empty night beds | Median | 0.13 | 0.13 | 0.13 | 0.13 | 0.11 |
| | IQR | 0.08 | 0.08 | 0.09 | 0.08 | 0.07 |
| | 5%–95% | 0.16 | 0.17 | 0.18 | 0.18 | 0.14 |
| Delayed transfers | Median | 0.73 | 0.77 | 0.91 | 1.04 | 1.36 |
| | IQR | 0.76 | 0.71 | 0.75 | 1.05 | 1.18 |
| | 5%–95% | 1.56 | 1.72 | 1.88 | 2.25 | 2.99 |
| Day bed:night bed ratio | Median | 0.10 | 0.10 | 0.10 | 0.11 | 0.11 |
| | IQR | 0.07 | 0.07 | 0.07 | 0.08 | 0.08 |
| | 5%–95% | 0.14 | 0.17 | 0.17 | 0.18 | 0.17 |
| ED conversion ratio | Median | 0.22 | 0.23 | 0.23 | 0.24 | 0.23 |
| | IQR | 0.09 | 0.09 | 0.10 | 0.09 | 0.08 |
| | 5%–95% | 0.19 | 0.19 | 0.19 | 0.19 | 0.19 |
| Admission casemix ratio | Median | 1.15 | 1.13 | 1.16 | 1.17 | 1.19 |
| | IQR | 0.39 | 0.51 | 0.45 | 0.41 | 0.40 |
| | 5%–95% | 1.12 | 1.21 | 1.59 | 1.55 | 1.67 |
| Emergency admission:bed ratio | Median | 4.42 | 4.47 | 4.62 | 4.76 | 4.93 |
| | IQR | 0.94 | 1.10 | 1.05 | 0.99 | 1.08 |
| | 5%–95% | 2.43 | 2.45 | 2.61 | 2.65 | 2.69 |

ED, emergency department.

cancelled elective operations. Between 2014 and 2016, the variables ED conversion ratio and admission casemix ratio show increasingly positive associations with cancelled elective operations. These variables also demonstrate a relatively high ability to explain the outcome variable in comparison with the other variables in the study ($R^2$ values 0.07 and 0.11, respectively, in 2016). The delayed transfers variable is observed to increase in importance between 2013 and 2016. The other variables included in the study did not show evidence of any clear association with cancelled elective operations.

### Cancelled electives: multivariate regression analysis

Table 6 shows a summary of the variables included in the multivariate linear regression model using cancelled elective operations as an explanatory variable in 2012 and 2016. In 2012, the only statistically significant variable related to cancelled elective operations was ED conversion ratio, which had a positive association. In 2016, the results indicate that the statistically significant variables in explaining variation were ED conversion ratio and admission casemix ratio. Both of these variables were observed to be statistically significant in the model between 2014 and 2016. Overall, the model was able to account for 17% of the variation in cancelled elective operations in 2016, which demonstrates an increasing ability to determine cancelled elective operations over the study period (increasing from 7% in 2012).

### DISCUSSION
### Summary of findings

The intrahospital patient flow variables with greatest association with better acute trust 4-hour target performance in England were found to have varied over the period 2012–2016. The main variables of interest in explaining performance in 2012 were found to be the proportion of empty night beds and ED conversion ratio. These variables were seen to have reducing importance over

**Table 3** Univariate regression for the breached attendances outcome variable

| Variable | Parameter | Year | | | | |
| | | 2012 | 2013 | 2014 | 2015 | 2016 |
|---|---|---|---|---|---|---|
| Empty day beds | $R^2$ | 0.02 | 0.01 | 0.01* | 0.00* | 0.00 |
| | Reg coef (95% CIs) | −0.001 (−0.003 to 0.001) | −0.001 (−0.004 to 0.002) | 0.001 (−0.002 to 0.005) | 0.001 (−0.003 to 0.006) | 0.002 (−0.004 to 0.008) |
| | P value | 0.15 | 0.43 | 0.4 | 0.54 | 0.6 |
| Empty night beds | $R^2$ | 0.10* | 0.09 | 0.07 | 0.05 | 0.06* |
| | Reg coef (95% CIs) | −0.012 (−0.019 to −0.006) | −0.015 (−0.022 to −0.007) | −0.017 (−0.027 to −0.006) | −0.017 (−0.031 to −0.004) | −0.025 (−0.042 to −0.008) |
| | P value | <0.01 | <0.01 | <0.01 | 0.01 | <0.01 |
| Delayed transfers | $R^2$ | 0.04* | 0.02 | 0.01* | 0.01 | 0.04 |
| | Reg coef (95% CIs) | 0.006 (0.001 to 0.011) | 0.005 (−0.001 to 0.011) | 0.003 (−0.004 to 0.011) | 0.003 (−0.005 to 0.011) | 0.011 (0.002 to 0.020) |
| | P value | 0.01 | 0.13 | 0.36 | 0.43 | 0.02 |
| Day bed:night bed ratio | $R^2$ | 0.03 | 0.02 | 0.01 | 0.04 | 0.05 |
| | Reg coef (95% CIs) | −0.058 (−0.118 to 0.002) | −0.066 (−0.138 to 0.006) | −0.058 (−0.156 to 0.040) | −0.146 (−0.264 to −0.027) | −0.218 (−0.378 to −0.058) |
| | P value | 0.06 | 0.07 | 0.25 | 0.02 | 0.01 |
| ED conversion ratio | $R^2$ | 0.07 | 0.05 | 0.05* | 0.05 | 0.03 |
| | Reg coef (95% CIs) | 0.071 (0.027 to 0.115) | 0.082 (0.025 to 0.140) | 0.114 (0.033 to 0.195) | 0.143 (0.039 to 0.248) | 0.144 (0.000 to 0.288) |
| | P value | <0.01 | 0.01 | 0.01 | 0.01 | 0.05 |
| Admission casemix ratio | $R^2$ | 0.00 | 0.00 | 0.00 | 0.00* | 0.00 |
| | Reg coef (95% CIs) | 0.002 (−0.007 to 0.010) | 0.001 (−0.010 to 0.012) | −0.002 (−0.016 to 0.012) | 0.007 (−0.012 to 0.025) | −0.001 (−0.027 to 0.024) |
| | P value | 0.70 | 0.86 | 0.77 | 0.47 | 0.93 |
| Emergency admission:bed ratio | $R^2$ | 0.01 | 0.02 | 0.01* | 0.04 | 0.01 |
| | Reg coef (95% CIs) | 0.002 (−0.001 to 0.006) | 0.003 (−0.001 to 0.008) | 0.003 (−0.002,0.009) | 0.009 (0.001 to 0.017) | 0.005 (−0.005 to 0.015) |
| | P value | 0.21 | 0.14 | 0.25 | 0.03 | 0.33 |

All results were based on data from 131 or more trusts.
*Indicates non-normality, heteroskedasticity or influential outlier in regression.
ED, emergency department; Reg coef, regression coefficient, β.

the subsequent years in the multivariate model. The day bed:night bed ratio variable was observed to have increased in importance between 2012 and 2016, and in the years 2015–2016, it was the most important patient flow factor in explaining 4-hour target performance in the multivariate model. The results also show that intra-hospital patient flow is only responsible for explaining some of the variation in 4-hour target performance, and

**Table 4** Multivariate regression model output for breached attendances as explanatory variable ($R^2$=0.19 in 2012, $R^2$=0.12 in 2016)

| Variable | 2012 | | | 2016 | | |
| | Regression coefficient | P value | 95% CIs | Regression coefficient | P value | 95% CIs |
|---|---|---|---|---|---|---|
| Empty night beds | −0.010 | <0.01 | −0.016 to −0.004 | −0.016 | 0.09 | −0.034 to 0.003 |
| Delayed transfers | 0.004 | 0.12 | −0.001 to 0.008 | 0.006 | 0.24 | −0.004 to 0.015 |
| Day bed:night bed ratio | −0.045 | 0.12 | −0.102 to 0.012 | −0.169 | 0.04 | −0.330 to −0.008 |
| ED conversion ratio | 0.068 | <0.01 | 0.025 to 0.112 | 0.104 | 0.17 | −0.044 to 0.252 |

ED, emergency department.

**Table 5** Univariate regression output for the cancelled elective operations outcome variable

| Variable | Parameters | Year 2012 | 2013 | 2014 | 2015 | 2016 |
|---|---|---|---|---|---|---|
| Empty day beds | $R^2$ | 0.00 | 0.00 | 0.00 | 0.00 | 0.00 |
| | Reg coef (95% CIs) | 0.009 (−0.047 to 0.066) | 0.009 (−0.046 to 0.064) | 0.008 (−0.051 to 0.066) | −0.014 (−0.079 to 0.051) | −0.003 (−0.069 to 0.062) |
| | P value | 0.75 | 0.75 | 0.8 | 0.68 | 0.92 |
| Empty night beds | $R^2$ | 0.01 | 0.02 | 0.00 | 0.00 | 0.02 |
| | Reg coef (95% CIs) | −0.135 (−0.328 to 0.059) | −0.130 (−0.307 to 0.047) | −0.087 (−0.273 to 0.099) | −0.066 (−0.260 to 0.129) | −0.149 (−0.336 to 0.039) |
| | P value | 0.17 | 0.15 | 0.36 | 0.51 | 0.12 |
| Delayed transfers | $R^2$ | 0.01 | 0.00* | 0.01* | 0.03* | 0.03 |
| | Reg coef (95% CIs) | 0.087 (−0.064 to 0.237) | 0.040 (−0.097 to 0.178) | 0.065 (−0.063 to 0.192) | 0.108 (−0.007 to 0.222) | 0.103 (0.004 to 0.201) |
| | P value | 0.26 | 0.56 | 0.32 | 0.07 | 0.04 |
| Day bed:night bed ratio | $R^2$ | 0.01 | 0.00 | 0.00 | 0.00 | 0.00 |
| | Reg coef (95% CIs) | 1.067 (−0.733 to 2.868) | 0.589 (−0.997 to 2.175) | 0.245 (−1.410 to 1.900) | 0.255 (−1.444 to 1.954) | −0.677 (−2.437 to 1.082) |
| | P value | 0.24 | 0.46 | 0.77 | 0.77 | 0.45 |
| ED conversion ratio | $R^2$ | 0.05 | 0.02 | 0.07 | 0.06 | 0.07 |
| | Reg coef (95% CIs) | 1.728 (0.402 to 3.055) | 1.020 (−0.288 to 2.327) | 2.275 (0.914 to 3.637) | 2.108 (0.627 to 3.589) | 2.419 (0.901 to 3.937) |
| | P value | 0.01 | 0.13 | <0.01 | <0.01 | <0.01 |
| Admission casemix ratio | $R^2$ | 0.02 | 0.05 | 0.13 | 0.12* | 0.11 |
| | Reg coef (95% CIs) | 0.240 (−0.016 to 0.496) | 0.318 (0.082 to 0.553) | 0.507 (0.288 to 0.726) | 0.524 (0.275 to 0.772) | 0.518 (0.258 to 0.778) |
| | P value | 0.07 | <0.01 | <0.001 | <0.001 | <0.001 |
| Emergency admission:bed ratio | $R^2$ | 0.00 | 0.00 | 0.00 | 0.00 | 0.00 |
| | Reg coef (95% CIs) | 0.029 (−0.077 to 0.136) | 0.010 (−0.087 to 0.108) | 0.009 (−0.089 to 0.107) | −0.025 (−0.136 to 0.087) | −0.033 (−0.143 to 0.077) |
| | P value | 0.58 | 0.84 | 0.85 | 0.66 | 0.56 |

*Indicates non-normality, heteroskedasticity or influential outlier in regression.
ED, emergency department; Reg coef, regression coefficient, β.

between 2012 and 2016 demonstrated a reducing ability to explain this variation ($R^2$ value of model reducing from 0.19 to 0.12). There was limited evidence of a clear association between delayed transfers of care and 4-hour target performance, either in the univariate or multivariate model results.

The main intrahospital patient flow variables associated with higher levels of cancelled elective operations at acute trusts between 2012 and 2016 were ED conversion ratio and the admission casemix ratio. Between 2012 and 2016, the importance of the admission casemix ratio is observed to have increased beyond that of ED conversion ratio.

**Table 6** Multivariate regression model output for cancelled electives as explanatory variable ($R^2$=0.07 in 2012; $R^2$=0.17 in 2016)

| Variable | 2012 Regression coefficient | P value | 95% CIs | 2016 Regression coefficient | P value | 95% CIs |
|---|---|---|---|---|---|---|
| Delayed transfers | 0.067 | 0.37 | −0.081 to 0.215 | 0.070 | 0.15 | −0.025 to 0.164 |
| ED conversion ratio | 1.625 | 0.02 | 0.302 to 2.949 | 1.881 | 0.01 | 0.385 to 3.377 |
| Admission casemix ratio | 0.225 | 0.08 | −0.027 to 0.477 | 0.483 | <0.001 | 0.230 to 0.736 |

ED, emergency department.

The ability of the multivariate model to explain variation in cancelled elective operations has increased between 2012 and 2016 ($R^2$ value of model increasing from 0.07 to 0.17), indicating that the emergency workload that trusts face are becoming an increasingly important factor in elective procedures being cancelled in England.

### How important is intrahospital flow to the 4-hour target?

The ability of the multivariate model to explain only 12% of the variation in 4-hour target in 2016 (which has reduced since 2012 from 17%) indicates that there are other factors that are increasingly important in determining 4-hour performance. By comparison, another multivariate model could predict only 6.8% variation in 4-hour target (using patient demographics and satisfaction with general practitioner (GP) service rates).[22] There are a number of factors that may affect ED processes and performance measures: macrohospital or intrahospital flow (ie, patient flows across the whole trust, such as those investigated in this study); microflow factors within departments/wards (ie, staffing,[4 17] work flows,[26 31] access to diagnostics[17]); population factors (ie, age, sex, deprivation,[22 32] access to GP/community/social care services[22 33]); noise (ie, recording errors, reporting differences,[10] 'gaming' of 4-hour target[5 17]). If the mechanisms of pressures facing acute trusts are to be understood more fully, future work is required to quantify the relative effects of each of these factors on trust performance measures.

### Bed occupancy

In our univariate analysis, night-bed occupancy was consistently associated with 4-hour performance (2012–2016). This is a result consistent with queuing theory. That is, if trusts do not provide adequate bed 'buffer' capacity to cope with the peaks of emergency admission demand, then ED performance will decline. As such, trusts must continue to focus on innovations to reduce night-bed occupancy. Targets such as 85% occupancy have been proposed elsewhere[34] although this seems unrealistic given in 2012 and 2016 respectively only 39% and 25% of trusts attained this level of night-bed occupancy (or lower).

### Day bed:night bed ratio and using day beds as 'buffers'

Our results show that that trusts with a higher ratio of day-bed to night-bed capacity were more likely to have higher 4-hour performance in 2016. It has been highlighted that admission and discharge patterns through trusts have peaks at different times of day.[9 11] Hence, an explanation for this result is that day-bed capacity can be used flexibly as a temporary 'buffer' for patient admission while the discharge of patients from the trust catches up during the day. Day beds are defined as consultant-led beds that are closed overnight,[28] hence patients are not able to occupy them for long periods as in the case of night beds, and the occupancy of such wards/areas will be low at the beginning of each day to allow admissions. It is possible that more trusts are taking advantage of this flexibility as the day bed:night bed ratio of trusts in England has increased between 2012 and 2016 (see table 2). This may be in a response to trusts operating at greater levels of occupancy (also see table 2) towards the end of this period and requiring a way of ensuring bed 'buffers' are available to allow patients to be admitted more promptly. We note that if occupancy can be reduced, this may remove the need to use day beds as internal buffers.

### Delayed transfers

Our expectations were that delayed transfers of care would be a strong predictor of 4-hour performance. Our results do not support this commonly held assumption. This suggests that hospital initiatives to reduce delayed transfers of care may not yield the expected benefits for ED waiting times. We in no way suggest that initiatives to reduce delayed transfers are not worthwhile, as expedited transfers to appropriate care such as rehabilitation or social care have clear clinical and quality of life benefits for patients. We do note the strong association between inpatient bed occupancy (night-bed occupancy) and delayed transfers of care (Spearman's rank correlation in 2016=−0.27, p=0.002). However, it is clear that a focus on only delayed transfers will not reduce occupancy to a level that is sufficient to release the pressure on EDs; more holistic approaches to reducing bed occupancy may be required.

### 'Clinical streaming' and conversion ratio

'Clinical streaming' in ED aims to triage patients within 15 min of their arrival and refer patients to other appropriate services.[35] This aims to reduce the load on EDs by only treating patients who cannot be treated by other services and provide better patient flow within EDs by prioritising different routes of care suited to patient needs. With regard to our study focusing on patient flow, it is plausible that clinical streaming may affect ED conversion ratio as there is currently some limited evidence that earlier review by senior clinicians may reduce avoidable admissions.[36] Our results show that ED conversion ratio was important in explaining some variation in 4-hour performance and cancelled elective operations; however, conversion ratios of trusts in England have not changed noticeably between 2012 and 2016 (see table 2) while some trusts are already known to have introduced clinical streaming.[17] It is currently unclear if there would be significant changes to trust conversion ratios with the proposed roll-out of clinical streaming[6] by NHS England. More research is needed to understand if clinical streaming impacts on patient flow in a positive manner.

### Strengths and weaknesses of study

This study is the first to examine in detail the in-hospital factors that influence ED 4-hour performance. We have analysed the change in the importance of common explanatory theories of hospital flow bottlenecks across 5 years using recent openly published data. We believe the

study provides a simple and transparent analysis that can contribute to the discussion around the causes of decline in ED 4-hour performance targets.

Trust 4-hour target reporting is known to be affected by 'gaming',[5] which may introduce extra variability into the relationships under investigation. However, this is the measure which trusts are judged and funded; therefore, it is argued that it is suitable for use within this study. The quality of data recording by trusts of the 'delayed transfers' variable has been reported to be questionable[10] and under-reported[37]; however, there is currently little published evidence around this issue. The data quality may explain the uncertainty in the importance of this variable over time in our analysis.

The $R^2$ values found, although higher than other multivariate models predicting the same outcome,[22] indicate that there is limited ability of the models to predict the outcome variables. It is possible that more complex statistical methodologies could provide a greater predictive capability. For example, it is plausible that extreme values in some of the variables investigated may lead to changes in another, and otherwise may have little impact. This may, for example, explain our results for delayed transfers of care. Hence, one possible example could be the use of generalised additive models, which would be able to account for possible non-linear relationships between the rates and ratios investigated in this study. It is also possible there may be trusts where specific variables impact on outcome measures but are not relevant to other trusts. It may be possible to use mixed models to investigate this further. A detailed longitudinal analysis could also be of benefit to provide greater understanding of the dependence of the variables over the study period. The data and open-source analysis code are supplied with this publication to allow the further development of this work and the open development of more complex models.

This study focusses on the macroflow factors across trusts. Future work including the relative importance of the macroflow, microflow, population factors and noise would be of value to assess the different pressures trusts face and aid in the targeting of service reconfigurations.

**Acknowledgements** We thank the NHS England Statistics Unify2 team for help in accessing and understanding the open data used in this study.

**Contributors** RG-L created the initial research question. BK contributed to the initial design of the work and was responsible for the protocol/ethics, data acquisition and collation, coding, analysis and interpretation of results. DC was responsible for the statistical methodology and interpretation of results. TM was involved throughout the study with guidance in the design, methodology and interpretation of results. All authors were involved in the development and review of the manuscript.

**Funding** This article presents independent research funded by the National Institute for Health Research (NIHR) Collaboration for Leadership in Applied Health Research and Care (CLAHRC) Wessex.

**Disclaimer** The views expressed in this publication are those of the author(s) and not necessarily those of the National Health Service, the NIHR or the Department of Health and Social Care.

**Competing interests** None declared.

**Patient consent** Not required.

**Ethics approval** This study was reviewed by University of Southampton ERGO ethics committee (reference 25957).

**Provenance and peer review** Not commissioned; externally peer reviewed.

**Data sharing statement** Data utilised in this study can be accessed via the Dryad data repository at doi:10.5061/dryad.n71684c. Further data associated with this study is available at doi:10.5258/SOTON/D0218.

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
