## [Reviewer comments · BMJ Open]

ARTICLE DETAILS

TITLE (PROVISIONAL)	Exploring emergency department four-hour target performance and cancelled elective operations. A regression analysis of routinely collected and openly reported NHS trust data.
AUTHORS	Keogh, Brad; Culliford, David; Guerrero-Luduena, Richard; Monks, T

VERSION 1 – REVIEW

REVIEWER	Mirza Mohamod Zahir Uddin Bhuiyan Mankweng hospital, Limpopo South Africa
REVIEW RETURNED	15-Nov-2017

GENERAL COMMENTS	1. Nothing mentioned about the elective operation 2. In the Title Authors mentioned 4 hour performance target in Emergency department and cancelled elective operation but there is nothing reflected about cancelled operation in results and conclusion 9. No, because results is focusing on 4 hours target, whereas one of the main outcome measured cancelled operation. It should be more detail about cancellation of elective operation 10. Statistics is very complex and it is not user friendly for general audience like doctors. It looks more for statistician and so many tables. Need to simply for general doctors. General comments • It is obvious that a lot of work was done; there is a sophisticated statistical attitude.• Key word A & E should be bracket after Accident & Emergency• Page 8 Table 4, line 74 R2 should be 0.12 as stated in text.• Page 11, line 137 , repeated that twice• Needs some English correction
--

REVIEWER	Adrian Boyle Addenbrookes Hospital, Cambridge
REVIEW RETURNED	01-Dec-2017

GENERAL COMMENTS	Thank you for asking me to review this interesting paper. I have some comments that I will hope will improve the usefulness of the manuscript. 1. The Methods section needs to specify the study design. This is basically a series of cross-sectional studies. The reference to the STROBE statement is irrelevant and should go. 2. I'm a bit concerned about which EDs they included. The Department of Health include Minor Injury Units, Eye Casualties, Urgent Care Centres and Walk In Centres in 'A&E' as Type 2 and Type 3 A&Es. Including these has a distorting and biasing effect. Firstly, they tend to have better four hour target performance, so this
---

	buffers any trend of declining performance (and this effect is not constant over the study period as the number of attendances at non Type 1 EDs has increased much more over the study period. Secondly, including these units will bias almost of the explanatory variables. I think the authors have to either 1. specifically exclude the non type 1 A&Es from their analysis and repeat the analysis or 2. Clarify better in the text that they have accounted and mitigated for this potential bias. 3. The authors need to discuss the limitations of the recording of the explanatory variables. Delayed transfers of care has been inconsistently recorded over the study period, and usually represents about a quarter to a third of all patients who are medically fit for discharge. This may explain their slightly counter-intuitive findings. 4. The discussion about Clinical Streaming isn't accurate (and obliquely relevant to this paper) Streaming reduces time to initial clinician contact, but doesn't reduce admissions or the conversion ratio (it may increase it)
--	---

REVIEWER	Robert M West University of Leeds, UK
REVIEW RETURNED	22-Dec-2017

GENERAL COMMENTS	The work attempts to answer an interesting problem and the analysis has the advantage of being simple and so accessible to many. My view though is that the authors have misjudged the sophistication required. The models are all very poor indeed as shown by the very low R2 values throughout. My view is that with such poor models, the interpretation is far from convincing. The univariate regressions simply formalise an inspection of linear correlation between two rates and find little correlation. So there is little to learn. The associations that have previously been published must be 'hidden' from this analysis because of the general assumptions made. The authors have explored assumptions of distribution of residuals graphically although graphs are not included. This is not the most important assumption though. There are three that stand out for me: (1) there are linear relationships between the rates. I find this not credible since extreme values of one may lead to changes in the other but otherwise there may be little influence. Note that the authors have also eliminated outliers, defined by Cook's distance, which could be very interesting in this respect. Generalise additive models might have improved modelling. (2) homogeneity of effect. Specifically that the effects in every hospital are the same. There may be hospitals where specific effects impact on 4-hour rates but are not relevant to other hospitals. A mixture model would have explored this situation and perhaps greatly improved the model. (3) The dependence of results has been circumvented by considering separate years. There would be dependence between years and this would have been extremely interesting in its own right. The authors have identified an important issue and data to explore it. The statistical models used are clearly extremely poor and need to be improved before valid interpretation can be achieved. Thus rather than reject, I recommend that the authors completely rework their analyses.
--

REVIEWER	Gichuru Phillip
-----------------	-----------------

	University of Central Lancashire Lancashire Clinical Trials Unit, UK
REVIEW RETURNED	28-Dec-2017

GENERAL COMMENTS	On the onset, after reading the abstract I understand that various NHS trust activity measures are routinely used to assess various performance outcomes which at the moment, presumably, are low. So it is not very clear whether this paper wanted “to explain the performance of various NHS trusts using these activity measures” or “to explain why (or show that) these activity measures are unable to explain (have changed) current NHS performance outcomes”. The authors seem to allude to the later in the discussion. The following is a summary of my take on the research reported. The various trust activity measures are counts, or ratios or aggregates of the same. Therefore;  1. What is the rationale for transforming the outcome variables? Is the log transformation done solely to normalise these potentially skewed data for purposes of fitting a regression model or is there another reason? e.g. if the outcome is normally distributed then various performance thresholds can be selected given NHS guidelines which may stipulate that a trust should keep its breached attendances below 5% or above 95%. In that case I can see why one would strive to get the outcome distribution to adhere to a normal distribution; the IQR and 5-95% ranges give an indication of the spread of the data, are both necessary? Does the term “Gradient” reported on tables refer to the Beta coefficient from the regression models? 2. What does the log transformation do in as far as interpretation of the model parameters i.e. does one now get the antilog of the computed model estimate while trying to predict the outcome of a given trust? 3. What about linearity assumption? 4. The authors explain that the variable “empty day beds” had zero counts thus a log transformation was not viable. While there are other regression methods to deal with such (zero) truncated data, we are not told the extent of this. When a trust has all beds full, this variable seems to lose its rationale since when there are no empty day beds it means the trust was really busy. Thus in this context and for purposes of this research is that a good thing or bad thing? Well, categorising the variable makes sense, but that also opens up the authors to justify their selection of bins which is not elaborated. 5. When the transformation was done, to what extent did it remove the skewness from the various variables, possibly a normality test was done or should be reported. 6. The normality or constant variance assumptions are made on the random error term in the regression model, and not on the outcome. Indeed if the outcome is skewed then it is likely that these assumptions will not be reflected on the error terms of the selected model but the motivation to transform for purposes of adherence to classical regression assumptions should stem from checking the distributions of the random error terms. 7. On missingness, the authors declare that a maximum of 4% was found for each variable. While this is low, and possibly its influence on the analysis not very impactful, the scenario changes if I have 2 variables in a model each with a 4% missingness. This is especially if the cases where the 2 variables are missing are not the same. So the claim of a maximum of 4% missing can only be made indeed the same cases (an outcome) of variables used in the model were missing, if this is not the case, then this missing case % is higher. 8. Further, such research is dependent on the use of NHS data
--

dashboards that collect and keep patient information. I do appreciate that collating these information and manipulate it to bring it to analysable quality and form can be a daunting task and to a large extent you can only work with what you have. However, are the authors sure that the updating of databases of various trusts is equally up to date? I could be wrong but to my knowledge some of these data bases are updated in a batch process; some of which is to some extent manual. Therefore if you have data for 2012 as of December 2012, the 2012 data will only be sufficiently up to date mid the following year after all the last quarter 2012 data has been updated.

9. The authors do well to start with a univariate analysis to select variables to be used for the multivariate regression. However, the selection or retention of variables to move to the next stage of model building is not very clear even if the authors note that clinical relevance heralds everything else. Aside from a variable showing considerable association strength to the outcome what other criteria, if any was used. I would expect that many of the trust activity variables are correlated positively or negatively, so were measures like variance influence factor assessed?

10. I would be weary about carrying out the regression by year (variation explained is quite low), but understand the authors avoiding that route of using longitudinal methods to keep the analysis simple and they do say at this time it is exploratory. In any case most of the time what an 'inferior' analysis concludes is just what the supposedly complex analysis will eventually allude to. However, by doing the analysis by year, I believe that the very essence of this work to show the change in performance over the years using various activity measures is lost. It leaves the reader to now process the yearly results and thus the whole "trend" analysis, which I think is the gist of this work is hidden. If the classical regression is kept, then maybe some form of baseline correction (for each quarter or year) should be done. Then perhaps some line graphs of how the various outcomes have changed over time would help to quickly give a snap shot of what has happened since 2012, unless these were left out due to journal space restrictions.

11. The change in time is important as it also shows how variable variables have changed over time possibly due to changes in policy within the NHS as discussed by the authors. It would be good to see a comparison of the various NHS trust characteristics vis a vis their expected patient traffic give time of the year; there are bound to be regional differences not only due to size or type of trusts, trust specific response to policy or social political changes, staffing but more importantly I think each counties trusts have a general working or patient management system that is being masked by pooling all data together. Therefore consider;

a. Somehow correcting for these differences in the modelling. However, I suppose that this is somehow reflected by variables like bed ratios and ED conversion ratios.

b. Keeping the same analysis germane, maybe separate small, medium or major injury analysis separately as outlined in line 138.

12. In the relation to point 10 above, were other methods of analysis considered? e.g. Multivariate analysis of variance or simultaneously using both outcomes together in one model if they are correlated i.e. multivariate regression? Both on the backbone of a repeated measures framework.

Even if I dwell of the negative aspects (some of which may stem from my lack of expertise and thorough context in ED patient flow), I do feel that this research really comes from a good place and the authors can be able to adequately assuage my concerns. Good luck.

VERSION 1 – AUTHOR RESPONSE

Reviewer: 1

Reviewer Name: Mirza Mohamod Zahir Uddin Bhuiyan Institution and Country: Mankweng hospital, Limpopo, South Africa Please state any competing interests or state 'None declared': None declared

	Reviewers comments	Author response
1	Nothing mentioned about the elective operation. In the Title Authors mentioned 4 hour performance target in Emergency department and cancelled elective operation but there is nothing reflected about cancelled operation in results and conclusion	We have amended the abstract to include results of cancelled elective operations. Cancelled elective operations are discussed in the Introduction, Methodology, Results and Discussion sections of the paper.
2	No, because results is focusing on 4 hours target, whereas one of the main outcome measured cancelled operation. It should be more detail about cancellation of elective operation	Please see the above response.
3	Statistics is very complex and it is not user friendly for general audience like doctors. It looks more for statistician and so many tables. Need to simply for general doctors.	We appreciate the reviewers request and the need to tailor the analysis for the audience. This is one of the primary motivations for not selecting a complex regression framework, such as a time series model, discussed by other reviewers. The linear regression framework chosen has been kept as simple as is reasonably possible. In order to provide a robust methodology aspects of the analysis, such as logarithmic transformation of variables, may be less intuitive, however is required for the underlying statistical methods to be valid.
4	General comments:  • It is obvious that a lot of work was done; there is a sophisticated statistical attitude. • Key word A & E should be bracket after Accident & Emergency • Page 8 Table 4, line 74 R2 should be 0.12 as stated in text. • Page 11, line 137 , repeated that twice • Needs some English correction 	We have addressed each of these comments where appropriate in the revised manuscript.

Reviewer: 2

Reviewer Name: Adrian Boyle

Institution and Country: Addenbrookes Hospital, Cambridge Please state any competing interests or state 'None declared': I have received research grants to investigate and mitigate emergency department crowding

	Reviewers comments	Author response
	Thank you for asking me to review this interesting paper. I have some comments that I will hope will improve the usefulness of the manuscript.	Thank you for these constructive comments. We feel they have allowed us to strengthen the paper.

1	The Methods section needs to specify the study design. This is basically a series of cross-sectional studies. The reference to the STROBE statement is irrelevant and should go.	We have now included the 'cross-sectional' study design in the abstract and methods sections. No standard reporting guidelines existed for our study methodology, however the STROBE RECORD framework seemed most appropriate. We have removed the reference to the guidelines within the manuscript text as per the reviewer's suggestion. We still submit the completed reporting guidelines as we feel it has strengthened the reporting of our study.
2	I'm a bit concerned about which EDs they included. The Department of Health include Minor Injury Units, Eye Casualties, Urgent Care Centres and Walk In Centres in 'A&E' as Type 2 and Type 3 A&Es. Including these has a distorting and biasing effect. Firstly, they tend to have better four hour target performance, so this buffers any trend of declining performance (and this effect is not constant over the study period as the number of attendances at non Type 1 EDs has increased much more over the study period. Secondly, including these units will bias almost of the explanatory variables. I think the authors have to either 1. specifically exclude the non type 1 A&Es from their analysis and repeat the analysis or 2. Clarify better in the text that they have accounted and mitigated for this potential bias.	As detailed in the section "Trust filtering" (p.5.) we have only included: Small, Medium, Large Acute and Teaching Trusts within the analysis (individual type 2 & 3 centres were removed). Within the included Trusts the activity is split into Type 1,2 &3 as the reviewer mentions. We agree with the reviewer that Type 2 & 3 A&E attendances included in a Trust's performance measures could have a biasing effect. We feel that the exclusion of Type 2&3 attendances could also have a biasing effect, particularly between Trusts which operate: Minor Injury Units, Eye Casualties, Urgent Care Centres and Walk In Centres at the same site as the Type 1 A&E (this could 'buffer the trend of declining performance' as suggested by the reviewer). It is therefore not clear that either method would be 'unbiased' in some manner. As this study is focussed on the overall patient flow across Trusts as a whole we felt the inclusion of all A&E workload across the Trust, regardless of type should be included. We feel this is an extremely valuable comment from the reviewer to include in the manuscript for clarity, and so we have done this in the "Bias" section (p.8).
3	The authors need to discuss the limitations of the recording of the explanatory variables. Delayed transfers of care has been inconsistently recorded over the study period, and usually represents about a quarter to a third of all patients who are medically fit for discharge. This may explain their slightly counter-intuitive findings.	There is currently little peer reviewed evidence, or even openly published reporting, around the issue. Given your comment, we have contacted the Nuffield trust and spoken to the Oak group (who are responsible for the data analysis with which the Nuffield trust have discussed the limitation in delayed transfer recording- to which presumably the reviewer refers), however their work is currently unpublished which makes a detailed discussion difficult. The limitations around Delayed transfers of care has been discussed and referenced where possible in the 'Strengths and Weaknesses of study' section p.14.
4	The discussion about Clinical Streaming isn't accurate (and obliquely relevant to this paper) Streaming reduces time to initial clinician contact, but doesn't reduce admissions or the conversion ratio (it may increase it).	This is an extremely helpful comment. We have reworked the discussion in section "'Clinical streaming' and conversion ratio" (p.14.) to focus on the possible impact clinical streaming would have on trust-wide patient flow; we believe the discussion is now much more relevant to the paper findings.

Reviewer: 3

Reviewer Name: Robert M West

Institution and Country: University of Leeds, UK Please state any competing interests or state 'None declared': None declared

	Reviewers comments	Author response
1	The work attempts to answer an interesting problem and the analysis has the advantage	We appreciate the reviewers concern. Our starting point for this analysis was that there has

	of being simple and so accessible to many. My view though is that the authors have misjudged the sophistication required. The models are all very poor indeed as shown by the very low R² values throughout. My view is that with such poor models, the interpretation is far from convincing. The univariate regressions simply formalise an inspection of linear correlation between two rates and find little correlation. So there is little to learn. The associations that have previously been published must be 'hidden' from this analysis because of the general assumptions made.	been limited publication in peer reviewed journals on the associations investigated in this work. A joint Health foundation /Nuffield trust report [ref 22] attempted to predict four-hour target performance using different explanatory variables and had an R² value (0.068) approximately 1/2 to 1/3 of the models we are presenting. The only peer reviewed reference [ref 12] similar to this work that we know of contained a single linear regression model for bed occupancy over a 2 weeks period in 2002. We hence see the aim of this paper as to provide a simple and transparent analysis that aims to determine if there are any basic associations between the multiple variables under investigation. We therefore believe that a formalised inspection of linear correlation of the variables is of use to a wide audience, and provides an initial study onto which more complex methodologies can then be applied, such as those suggested by the reviewer. We agree with the reviewer that there is little to learn in terms of using the models for performance prediction. However, the results (including the low R² values) are useful in providing learning and discussion around the possible relationships between the variables being investigated and how they have changed over time. These variables are commonly stated as being related in non-peer-reviewed literature, and so we attempt here to provide the data and initial analysis framework to start a body of work which can attempt to verify these statements. In response re: low R² values we have revised the manuscript and discussed this in more detail (section: "How important is intra-hospital flow to the four-hour target?" p. 13) and included a discussion around this point the review makes (section "Strengths and weaknesses of study" p.14) .
2	The authors have explored assumptions of distribution of residuals graphically although graphs are not included. This is not the most important assumption though. There are three that stand out for me:	The distributions of residuals have not been presented graphically (neither have the regression models themselves) in the interests of brevity within the paper. Some examples of each are now provided in an online supplementary Appendix A, we suggest only providing a selection as there are 70 univariate regressions. The data and (open source) analysis code are supplied online with the paper which can be run by any reader to examine the residual plots.
3	(1) there are linear relationships between the rates. I find this not credible since extreme values of one may lead to changes in the other but otherwise there may be little influence. Note that the authors have also eliminated outliers, defined by Cook's	Outliers were not removed in the analysis, they were checked for and flagged in the results tables. We have clarified our wording in the methodology section to make this clearer (see p.8). We accept generalised additive models (GAMs)

	distance, which could be very interesting in this respect. Generalise additive models might have improved modelling.	have the ability to deal with non-linear effects. As explained in response #1 above we are at the early stage of analysis, and we are presenting high level results to give a very general indication of whether there are relationships between the outcome and explanatory variables. We are not sure the results of a GAMs analysis would be interpretable to the general audience we are attempting to reach with this study, hence we do not feel that GAMs would be a suitable approach at this stage. This is an extremely valid point however and so we have added a discussion in the 'Strengths and weaknesses of study' section (p.14.) to discuss the possible non-linearity and possible use of other models in future work.
4	(2) homogeneity of effect. Specifically that the effects in every hospital are the same. There may be hospitals where specific effects impact on 4-hour rates but are not relevant to other hospitals. A mixture model would have explored this situation and perhaps greatly improved the model.	It is not clear to us how using a mixture model would solve the problem with effect homogeneity. Yes, a mixture model could be a better fit for the residual error distribution, allowing (for example) for a bi-modal distribution. However our aim with this paper is to estimate/describe the basic signals provided by a small range of specific explanatory variables, and we are reluctant to add a large amount of methodological complexity at this stage. It is possible the reviewer meant mixed models. We have attempted to build some mixed models in response to this. In our present analysis (by year) we are not able to conduct mixed models (random intercept model) by hospital as there is only one value per hospital (or 4 if using quarterly data- which is the highest granularity available across all variables). There is no obvious way to know how to group hospitals other than by 'trust type' i.e. large acute, small acute, teaching etc. However, this grouping leaves us with only 4 groups with which we are uneasy about applying a mixed model. Another option we have explored is to use the longitudinal quarterly data and to build a mixed model for each individual trust. However, there are then possible issues around autocorrelation which would need to be addressed in some manner. From our analysis it appears that mixed models could provide some interesting insight into the data however it is not obvious to us that this approach is not also without its limitations. We have included a discussion around this valuable point the reviewer raises in section 'Strengths and weaknesses of study'.p.14.
5	(3) The dependence of results has been circumvented by considering separate years. There would be dependence between years and this would have been extremely interesting in its own right.	We agree that a longitudinal study would be of great interest. With the data available online time series analyses may be conducted to further the initial work presented in our manuscript.
6	The authors have identified an important issue and data to explore it. The statistical models used are clearly extremely poor and need to be improved before valid	We thank the reviewer for their comments and their suggestions around using different statistical methodologies, which we have discussed in length and detail between co-authors. These

	interpretation can be achieved. Thus rather than reject, I recommend that the authors completely rework their analyses.	comments have helped us consider and scrutinise how else we may have approached the analysis of this data and how we may continue the analysis further. The data is structured such that it is very difficult to use some of these methodologies as the primary analysis tools. Applying some of these methodologies also appears to come with their own limitations. Please see the above responses for more a more detailed discussion around these points. We have acknowledged each of these points in detail in the manuscript in the 'Strengths and weaknesses of study' section p.14. We have also acknowledged the limitations of the conclusions we may draw from this analysis within the initial bullet points 'Strengths and limitations of this study' section following the abstract. We welcome the use of other models to investigate the aspects raised by the reviewer, to further build on this initial study. The data is available online for anyone to utilise to explore this further.
--	--	---

Reviewer: 4

Reviewer Name: Gichuru Phillip

Institution and Country: University of Central Lancashire, Lancashire Clinical Trials Unit, UK Please state any competing interests or state 'None declared': Non

	Reviewers comments	Author response
	On the onset, after reading the abstract I understand that various NHS trust activity measures are routinely used to assess various performance outcomes which at the moment, presumably, are low. So it is not very clear whether this paper wanted "to explain the performance of various NHS trusts using these activity measures" or "to explain why (or show that) these activity measures are unable to explain (have changed) current NHS performance outcomes". The authors seem to allude to the later in the discussion.	Our starting point for this analysis was that there is only a very limited amount of quantitative and transparent analyses that have been conducted exploring ED and hospital performance. In fact most evidence is not peer reviewed and informal. The objective of the paper is therefore "To quantify the effect of intra-hospital patient flow on Emergency Department (ED) performance targets and indicate if the expectations set by the NHS England five year forward review are realistic in returning emergency services to previous performance levels." We have stated this in the 'objectives' section of the abstract.
1	The various trust activity measures are counts, or ratios or aggregates of the same. Therefore; What is the rationale for transforming the outcome variables? Is the log transformation done solely to normalise these potentially skewed data for purposes of fitting a regression model or is there another reason? e.g. if the outcome is normally distributed then various performance thresholds can be selected given NHS guidelines which may stipulate that a trust should keep its breached attendances below 5% or above 95%. In that case	Yes, variables were transformed to provide normal distribution and hence the application of linear regression models. We have clarified this in the manuscript (p.8). We think the 5-95% ranges are important to present, as they can indicate in more detail the spread of the data. Yes, gradient refers to Regression Coefficient. After reviewing other

	I can see why one would strive to get the outcome distribution to adhere to a normal distribution; the IQR and 5-95% ranges give an indication of the spread of the data, are both necessary? Does the term “Gradient” reported on tables refer to the Beta coefficient from the regression models?	manuscripts published in the same journal we see that ‘regression coefficient’ is commonly used, we have amended our manuscript to match this. Thank you for helping us clarify.
2	What does the log transformation do in as far as interpretation of the model parameters i.e. does one now get the antilog of the computed model estimate while trying to predict the outcome of a given trust?	If one wishes to predict trust performance then antilog can be calculated. The prediction of trust performance is not the objective of this paper and hence we have not discussed this in the manuscript. In order to understand this further we refer the reviewer to: Bland and Altman, “Statistics notes: Transformations, means, and confidence intervals”, BMJ 1996;312:1079.
3	What about the linearity assumption? The authors explain that the variable “empty day beds” had zero counts thus a log transformation was not viable. While there are other regression methods to deal with such (zero) truncated data, we are not told the extent of this. When a trust has all beds full, this variable seems to lose its rationale since when there are no empty day beds it means the trust was really busy. Thus in this context and for purposes of this research is that a good thing or bad thing? Well, categorising the variable makes sense, but that also opens up the authors to justify their selection of bins which is not elaborated.	We have added a discussion in the “Strengths and weaknesses of study” section (p.14.) around the linearity assumption. We agree with the reviewer that categorising the variable “empty day beds” makes sense. We have now elaborated on the method for the creation of bins (“variable distributions and transformation” section, p.8.). Our method was chosen as a pragmatic way of categorising the variable.
4	When the transformation was done, to what extent did it remove the skewness from the various variables, possibly a normality test was done or should be reported.	In our methodology we visually inspect the variables using histograms. A normality test such as a Shapiro Wilk is not usually a standard requirement in medical statistics applications.
5	The normality or constant variance assumptions are made on the random error term in the regression model, and not on the outcome. Indeed if the outcome is skewed then it is likely that these assumptions will not be reflected on the error terms of the selected model but the motivation to transform for purposes of adherence to classical regression assumptions should stem from checking the distributions of the random error terms.	The transformations were applied in order to provide closer adherence of the random error term in the regression model. The “Statistical methods” section (p.8) explains: the univariate and multivariate models’ residuals were checked visually for normality and homogeneity. Abnormalities are reported in the results section.
6	On missingness, the authors declare that a maximum of 4% was found for each variable. While this is low, and possibly its influence on the analysis not very impactful, the scenario changes if I have 2 variables in a model each with a 4% missingness. This is especially if the cases where the 2 variables are missing are not the same. So the claim of a maximum of 4% missing can only be made indeed the same cases (an outcome) of variables used in the model were missing, if this is not the case, then this missing case % is higher.	This is an important point to clarify and we thank the reviewer for prompting us to do so. For each year studied no more than 4% of trusts were found to have missing data for at least one variable; therefore the maximum percentage of missing data points for any regression was less than 4%. We have clarified this in the ‘bias’ section (p.8.) to prevent confusion.
7	Further, such research is dependent on the use of NHS data dashboards that collect and keep patient information. I do appreciate that collating these information and manipulate it to bring it to analysable quality and form can be a daunting	The data is collated by NHS England and released in one ‘batch’. This is usually several months behind as the reviewer suggests. The full definitions and process that NHS England follows are referenced in

	task and to a large extent you can only work with what you have. However, are the authors sure that the updating of databases of various trusts is equally up to date? I could be wrong but to my knowledge some of these data bases are updated in a batch process; some of which is to some extent manual. Therefore if you have data for 2012 as of December 2012, the 2012 data will only be sufficiently up to date mid the following year after all the last quarter 2012 data has been updated.	the “Data collation” section p.5.
8	The authors do well to start with a univariate analysis to select variables to be used for the multivariate regression. However, the selection or retention of variables to move to the next stage of model building is not very clear even if the authors note that clinical relevance heralds everything else. Aside from a variable showing considerable association strength to the outcome what other criteria, if any was used. I would expect that many of the trust activity variables are correlated positively or negatively, so were measures like variance influence factor assessed?	We used a pragmatic approach to variable selection which included using variables highlighted previously as of clinical importance and additionally those which showed considerable association strength to the outcome. This has been more clearly stated in the manuscript in section: “Statistical methods” p.8.
9	I would be weary about carrying out the regression by year (variation explained is quite low), but understand the authors avoiding that route of using longitudinal methods to keep the analysis simple and they do say at this time it is exploratory. In any case most of the time what an ‘inferior’ analysis concludes is just what the supposedly complex analysis will eventually allude to. However, by doing the analysis by year, I believe that the very essence of this work to show the change in performance over the years using various activity measures is lost. It leaves the reader to now process the yearly results and thus the whole “trend” analysis, which I think is the gist of this work is hidden. If the classical regression is kept, then maybe some form of baseline correction (for each quarter or year) should be done. Then perhaps some line graphs of how the various outcomes have changed over time would help to quickly give a snap shot of what has happened since 2012, unless these were left out due to journal space restrictions.	As the reviewer suggests the decision to conduct classical regression and not more complex longitudinal analysis was made in order to keep the paper readable to as large a general audience as possible. We believe the use of a longitudinal approach would be counter to this objective at this stage. We do agree that there is potential for longitudinal analyses, however we see this as follow on work to the present study - which could be conducted by ourselves or the larger community. If readers are interested in time-series analysis we welcome the use of more complex models to investigate these aspects. The data is available online and referenced in the manuscript for anyone to use to explore these ideas further. With the lack of other peer-reviewed literature completing similar analyses the ones used in this manuscript we believe a simple analysis still provides a useful contribution, which may be built on further in future. We have added a discussion around this valuable point in the ‘Strengths and weaknesses of study’ section p.14. With regard to a baseline correction we wished to keep the analysis as transparent and easy to follow as possible, we are currently unsure of a suitable commonly used methodology to use in this regard. We feel that the inclusion of time series plots (which are already available online and in many of the reports referenced in the manuscript introduction) will not add value to the paper. In the interests of brevity and

		clarity of narrative in the manuscript these were not included.
10	The change in time is important as it also shows how variable variables have changed over time possibly due to changes in policy within the NHS as discussed by the authors. It would be good to see a comparison of the various NHS trust characteristics vis a vis their expected patient traffic give time of the year; there are bound to be regional differences not only due to size or type of trusts, trust specific response to policy or social political changes, staffing but more importantly I think each counties trusts have a general working or patient management system that is being masked by pooling all data together. Therefore consider; a. Somehow correcting for these differences in the modelling. However, I suppose that this is somehow reflected by variables like bed ratios and ED conversion ratios. b. Keeping the same analysis germane, maybe separate small, medium or major injury analysis separately as outlined in line 138.	We agree with the reviewer that the type of hospital and regional co-ordination of acute services may have an impact on hospital performance. We feel that the inclusion of this further analysis will lengthen the paper. As mentioned in reviewer comment 9: we welcome the use of more-complex models to investigate these aspects. The data is available online and referenced in the manuscript for anyone to use to explore these ideas further.
11	In the relation to point 10 above, were other methods of analysis considered? e.g. Multivariate analysis of variance or simultaneously using both outcomes together in one model if they are correlated i.e. multivariate regression? Both on the backbone of a repeated measures framework. Even if I dwell of the negative aspects (some of which may stem from my lack of expertise and thorough context in ED patient flow), I do feel that this research really comes from a good place and the authors can be able to adequately assuage my concerns. Good luck.	This is an interesting idea. No we did not consider a multivariate analysis of variance due to the difficulty explaining and interpreting the results from a multivariate (not-multivariable) regression model. Please see our response above to points 9 and 10. Many thanks for your extensive review and helpful comments.

VERSION 2 – REVIEW

REVIEWER	Gichuru Phillip University of Central Lancashire, Lancashire Clinical Trials Unit, UK
REVIEW RETURNED	01-Mar-2018

GENERAL COMMENTS	Dear Authors, Many thanks for review responses to various questions/suggestions I raised in the first review. I just have one more query on page 8 (on line 33) regarding influential outliers. After flagging potential outliers using Cook's distance, you made a decision not to remove these cases. Presumably after you investigated that the data was 'correct'; how many were these cases vis a vis your sample size? Then did you redo the analyses without these cases just to see if anything else changes (e.g. overall model fits, change in estimates, adherence to model assumptions, necessity to transform or even a slight change in significance of variables that may now even shake up your choice
---

	of variables in the model). Other than this am happy with the authors responses. All the best.
--	--

REVIEWER	Robert M West University of Leeds UK
REVIEW RETURNED	02-Mar-2018

GENERAL COMMENTS	I appreciate the responses that the authors have provide to my comments and queries. The corrections to the manuscript are sufficient to clearly state the reasons for limited statistical analysis at this stage. Therefore I strongly recommend publication.
--

REVIEWER	Bhuiyan MMZU University of Limpopo South Africa
REVIEW RETURNED	03-Mar-2018

GENERAL COMMENTS	Comments: I went through the paper again; a lot of things have been added and removed especially in the abstract. concern of elective cancellation of operation was addressed, and deserves to be published. But few things need to address 1. Some minor corrections and should be attended to before the paper goes for publications a. A & E is not a keyword but the abbreviation of Accident and Emergency b. Please attend to page 11 line 94 $R^2 = 0,12$ not 0,1 as stated in the text • The paper is not required to send to me after the minor revision, the editor will make sure that the corrections were inserted
---

VERSION 2 – AUTHOR RESPONSE

Reviewer: 4

Many thanks for review responses to various questions/suggestions I raised in the first review. I just have one more query on page 8 (on line 33) regarding influential outliers. After flagging potential outliers using Cook's distance, you made a decision not to remove these cases. Presumably after you investigated that the data was 'correct'; how many were these cases vis a vis your sample size? Then did you redo the analyses without these cases just to see if anything else changes (e.g. overall model fits, change in estimates, adherence to model assumptions, necessity to transform or even a slight change in significance of variables that may now even shake up your choice of variables in the model).

AUTHOR RESPONSE:

We did not remove influential outliers as the data used are those directly reported by the Trusts. We do not believe it appropriate for us to make assumptions about certain data being 'incorrect' and removing specific points. The methodology we have followed regarding outliers is clearly stated in the paper and the code/data are made available so that any reader wishing to complete a sensitivity analysis, such as the one outlined by Reviewer 4 above, is able to do so.

Reviewer: 1

I went through the paper again; a lot of things have been added and removed especially in the abstract.

concern of elective cancellation of operation was addressed, and deserves to be published. But few things need to address

1. Some minor corrections and should be attended to before the paper goes for publications
 - a. A & E is not a keyword but the abbreviation of Accident and Emergency
 - b. Please attend to page 11 line 94 $R^2 = 0,12$ not 0,1 as stated in the text
- The paper is not required to send to me after the minor revision, the editor will make sure that the corrections were inserted

AUTHOR RESPONSE:

- a. We have changed the keyword as suggested
- b. the R^2 has been corrected to 0.12 as suggested, to match the inline text. (see p11, L 96).